# An Online Rail Track Fastener Classification System Based on YOLO Models

**DOI:** 10.3390/s22249970

**Published:** 2022-12-17

**Authors:** Chen-Chiung Hsieh, Ti-Yun Hsu, Wei-Hsin Huang

**Affiliations:** 1Department of Computer Science and Engineering, Tatung University, Taipei 104, Taiwan; 2The Graduate Institute of Design Science, Tatung University, Taipei 104, Taiwan

**Keywords:** railway track inspection, deep learning, object detection neural network, YOLO, real time

## Abstract

In order to save manpower on rail track inspection, computer vision-based methodologies are developed. We propose utilizing the YOLOv4-Tiny neural network to identify track defects in real time. There are ten defects covering fasteners, rail surfaces, and sleepers from the upward and six defects about the rail waist from the sideward. The proposed real-time inspection system includes a high-performance notebook, two sports cameras, and three parallel processes. The hardware is mounted on a flat cart running at 30 km/h. The inspection results about the abnormal track components could be queried by defective type, time, and the rail hectometer stake. In the experiments, data augmentation by a Cycle Generative Adversarial Network (GAN) is used to increase the dataset. The number of images is 3800 on the upward and 967 on the sideward. Five object detection neural network models—YOLOv4, YOLOv4-Tiny, YOLOX-Tiny, SSD512, and SSD300—were tested. The YOLOv4-Tiny model with 150 FPS is selected as the recognition kernel, as it achieved 91.7%, 92%, and 91% for the mAP, precision, and recall of the defective track components from the upward, respectively. The mAP, precision, and recall of the defective track components from the sideward are 99.16%, 96%, and 94%, respectively.

## 1. Introduction

A railway comprises ballasts, sleepers, rails, and fasteners. Each component plays an important role, among which the rail fasteners can fasten the rail tracks to the sleepers and have a buffering effect to disperse the weight, which can slow down the disturbance of the rails so that the rails can be stably fixed on the rail tracks. Damaged or dropped fasteners may derail the train, so track inspection is among the essential work items.

Current track inspection is conducted manually and divided into foot and vehicle inspections. The inspection personnel inspect the track visually and record any damage in paper form, then send maintenance personnel to fix it. In May 2021, the Taiwan Railway Administration (TRA) used a mobile phone APP [1] to replace the traditional paper record to enhance the efficiency and reliability of rail route inspection. The digitization of inspection data reduces the likelihood of records being lost. However, it does not prevent human errors, and only the first version of Android has been developed for track inspectors to test.

Moreover, because the human eye way inspection is limited to the human visual inspection angle, inspection speed cannot be too fast so that the naked eye can see clearly. During foot inspection, the person must also beware of passing trains to avoid collisions. The cost of in terms of manpower and time is very high to complete inspection. Given the current workforce of the railway station, inspection can only be performed once a week. If the frequency of inspection is increased, this may cause the inspection staff to be overworked and even less able to inspect carefully. This also makes the transportation and security of the TRA impossible to be effectively refine. In order to improve the efficiency of patrol inspection, a real-time identification system for defective track components is developed based on computer vision to improve the inspection problem effectively.

In recent years, there has been much research on detecting track defects. We want to use artificial intelligence to carry out automatic detection, focusing on improving accuracy and real-time computing. If the recognition rate and stability can be higher than manual inspection, it can replace the current manual inspection to improve the efficiency of rail inspection. By comparing the current commercial patrol equipment products, Table 1 shows that most of them are measuring geometric property of track and components when the types of track components are pretty diverse. Moreover, because the whole equipment is quite expensive, only a very few copies of the systems can be purchased for a track operator. For example, there is only one automatic track inspection system running for Taipei Mass Rapid Transit and one track image recording system for the TRA. Due to the limited budget, there are few inspection vehicles, resulting in slow inspection efficiency.

An automatic detection system is emerging using computer vision technology to overcome the effort spent on human manual inspection of track defects. Two approaches have been developed for this task: traditional pattern recognition [7] and deep neural networks [8,9] based. The former approach first identifies the features that effectively classify defective track components from standard track components. Additionally, then a recognition engine was developed such as a classification tree [10], k-Near Neighbor (k-NN) [11], Support Vector Machine (SVM) [12], and neural network (NN) [13]. Recently, the deep learning-based approach has gained much attention for the impressive advance in object recognition and object detection. With the rapid computation provided by the graphic processing unit, the time spent is significantly reduced and could be deployed in real-world applications. Therefore, this work adopts the deep learning-based approach for rail track safety inspection.

Training deep learning neural networks needs a certain number of datasets to produce good identification results. However, the defective component images are not easy to obtain, so the number of images for the seriously damaged components can only be limited to less than ten. Therefore, this study needs to overcome the difficulties in data collection. Data augmentation methods such as resizing, rotation, and translation could be used to enlarge the dataset to obtain possible images in the actual situation. This research will use the manual treatment to generate the cracks on the regular rail by repairing the image, but the manual treatment can only produce a small amount of data. Thus, a Generative Adversarial Network (GAN) [14] is used to automatically generate the rail fracture dataset, as far as possible, to expand the data and to achieve good training results.

In this study, we want to develop a real-time identification system for rail defect components with good identification accuracy and relatively low cost using object detection and identification methods. GoPro (It is based in San Mateo, CA, USA. GoPro, Inc. Woodman Labs, Inc. U.S.) Hero8 Black and GoPro Hero7 Black motion cameras are used to capture images. Motion cameras have sound shock-resistant processing and high frames per second (FPS). Then, deep learning-based object detection is trained to identify defective track fasteners automatically. This study divides rail inspection into two modes: overhead and side inspection. The two models are trained separately. The complete fastener defect categories are also built from the two perspectives. Using the multi-thread technique, the computer can process the image transmitted by two cameras in real time. Our goal is to develop an automatic real-time identification system that can replace the present manual inspection, keep a reasonable accuracy rate to reduce the cost of inspection staff effectively, and improve the overall efficiency of inspection.

This study is divided into five sections. This section is an introduction to the motivation and background of the research and the research objectives and difficulties. The Section 2 is the literature review, divided into three parts. The first part introduces the current situation of rail track inspection; the second part will compare the differences between this study and other related studies; the third part introduces the AI identification model used in this research. In the Section 3, the system architecture is described in detail. Firstly, the system is designed and constructed, and includes a real-time defective component identification neural network model, two high-speed cameras, a GPS inertial navigation device, and a flat patrol car. The number of classes for the regular and defective rail components and the collected datasets for each class are described. The experimental results and analysis are presented in detail in the Section 4. The identification results of different neural networks are compared, and field tests verify the feasibility of the whole system. The Section 5 is the conclusion and prospects of this study.

## 2. Related Works

The vehicle wheels can apply vertical, lateral, and creep loads to the rails, while bulk stress, such as bending stress, thermal stress, and residual stresses, can also be applied to the rails. Different rail defects such as rail corrugations, rolling contact fatigue defects, squat defects, shatter cracking, split head, and wheel burns have their causes and characteristics, lead to different effects, and thus require corresponding treatments [15]. Defects and deteriorated conditions on the rail track can normally be seen by human eyes; therefore, manual inspection by patrollers can identify and locate the defects and monitor the condition. However, such inspections are labor intensive and can only be arranged during non-operating hours in order not to disrupt the regular service operations. Rail inspection vehicle and sensor technologies are being deployed as an efficient and cost-effective data collection technology solution to support rail maintenance operations and can capture vast amounts of data. Data-driven automatic condition monitoring and detection and classification of rail track anomalies have been attracting attention from researchers at universities and railway institutes.

Ma et al. [16] proposed a switched median filter coupled with improved Canny edge detection to extract the features of fastener edges. Due to the fastener’s fixed shape, the defective fastener’s contour differs from that of the standard fastener and the real-time missing identification could be carried out. The final fastener identification rate was 85.8%, and the average processing time was 245.61 ms. Gibert et al. [17] proposed a method for fastener detection by (1) carefully aligning the training data, (2) reducing intra-class variation, and (3) bootstrapping complex samples to improve the classification margin. The system can inspect ties for missing or defective rail fasteners using the histogram of oriented gradient as features and classified by a combination of linear SVM classifiers. The detection rate is 98%, and the false alarm rate of 1.23% on a new dataset of 85 miles of concrete tie images collected in the US Northeast Corridor.

Stella et al. [18] used wavelet transform and principal component analysis for fastener image pre-processing and employed a neural network classifier to detect defective hook-shaped fasteners. The AdaBoost algorithm was used by Li et al. [7] and Xia et al. [19] for detecting fasteners from rail images. In recent years, the application of deep learning methods for rail track inspection has gained significant importance due to the increase in computing power and the development of graphical processing units (GPU). Deep learning methods are producing success in various applications with recent technological advances. Deep learning has neural networks as its functional unit to mimic how the human brain solves complex problems based on data. Convolutional neural networks (CNN) [20] and object detection neural networks such as You Only Look Once (YOLO) [21] propelled the development of deep learning, which has been reported with convincing performances for rail track monitoring and anomaly detection. The performances in the prediction and learning of these methods are improving with the increasing amount of data available [22].

According to the review on deep learning approaches for rail track condition monitoring [9], the authors find research works published from 2013 to 2021. In total, they identified 62 relevant research publications to review. The trend over time indicates that the rail industries are adopting deep learning methods with growing interests. As for the detection/localizing/classification targets, it is observed that rail surface defects including various components (rail, insulator, valves, fasteners, switches, track intrusions, etc.) are the most common items. As for the deployed deep learning models, many deep learning models are adopted by researchers. Ji et al. [9] summarizes the distribution of deep learning models. CNN is the most popular one being adopted that results using ResNet [23,24] and LSTM [25]; however, many researchers created their own structure [26] or divided their tasks into a few stages such as SSD + VGG [27] and Faster RCNN + CNN [28]. Some deep learning methods just adopted one-stage object detection such as YOLOv2 [29], YOLOv3 [30], and YOLOv5 [31]. The effectiveness and the results differ from each other depending on the tasks.

Various deep learning methods are reported to produce promising results for rail track condition monitoring. However, there is a consistent process flow [9] for these deep learning methods. First, how to capture the raw image is the most critical step, which requires the installation of cameras/recording devices on rail maintenance vehicles. Second, the raw image data are transmitted to the image processing subsystem for pre-processing such as resizing, noise removal, or enhancement. Third, the collected images are labeled accordingly by the experts of rail workers to build the dataset. Fourth, the selected deep learning model is trained with the randomly selected training data and validated by the testing data. Depending on the selected strategy, a one-stage deep learning model could perform classification directly, or multi-stage deep learning methods require localization tasks for track components and then a classification task. It is also possible to perform localization and classification concurrently. Fifth, the trained deep learning model is put in operation with the trained parameters for real-world applications. Due to the criticality of rail track inspection, double checking the recorded rail track images by human operators is necessary to confirm the system accuracy. Finally, the efficiency and effectiveness of the deep learning models are reviewed and enhanced for improved performances.

Still, there is a key problem that needs to be addressed—the lack of images of the defective rail track component. The nature of rail operations causes the data distribution of defective rail track to be disproportionate, which could cause class imbalance problems in deep learning applications. Data augmentation is often used to produce more training data by resizing, rotating, and translating the original dataset. However, some gaps exist between the real image and the augmented one.

In this research, we follow the general process flow to construct a real-time online rail track inspection system by the one-stage approach. Online means the captured image is immediately transmitted to the recognition engine for defective component detection, unlike offline recognition, where the recognition engine works at the backend server. The recorded video is sent to the server when the whole rail segment is finished checking. The significant contribution of our research work is that we encapsulate the recognition engine as a parallel running process waiting to process received images. The second contribution is using the latest version of object detection models YOLOv4 and YOLOX for one-stage detection of defective components. Due to the lack of image data of defective components, the third contribution is using GAN for data augmentation, which could generate image data close to real ones. The last contribution but not the least, is the classification of defective rail components into two categories. There are ten defects covering fasteners, rail surfaces, and sleepers from the upward and six defects about the rail waist and fishplate from the sideward.

## 3. Proposed Methodology

Owing to the significant advance of deep learning-based object detection, we adopt the one-stage approach to detect defective rail components directly. Thus, we first describe the deep learning-based object detection models and then the proposed system architecture for detecting defective rail components. To cover most parts of rail components, we divide these items according to the camera viewing angles. One is from the top view, and the other is from the side view. Those classes are defined by rail experts and summarized in the last subsection.

### 3.1. Object Detection Neural Network Models

Object detection uses “Box” in images such as photos or videos to mark the scope of an object and classify it. Object detection can be divided into two broad categories: two stage and one stage. Two stage refers to selecting and identifying objects, while one stage refers to a neural network that can simultaneously detect and classify objects. This method is fast, and the identification rate is still in a specific acceptance range. The object identification models of one stage such as YOLO and SSD are compared in this study.

#### 3.1.1. YOLOv4

YOLO [21] is a one-stage object detection method. Joseph Redmon first proposed it in June 2015, and then Alexey Bochkovskiy proposed the YOLOv4 [32] neural network in 2020. This neural network is mainly used for object detection. YOLOv4 can not only classify objects but also frame the position of objects. YOLOv4 has made many changes in various parts of YOLOv3 [33]. At the same time, the identification speed is guaranteed, the identification precision of the whole model is improved, and the hardware specification requirements of identification and training are reduced.

YOLOv4’s main contribution is to build an efficient and powerful object detection model. YOLOv4 can be trained using a single GPU. Backbone uses CSP Dark Net53 as its pre-training backbone. In the convolution layer, a few groups (1~8 groups) of convolutions are used to lighten the model and reduce the parameters significantly as well as reduce the overall computational load and shorten the forecast time. Neck uses SPP + PAN. In contrast to YOLOv2, the SPP will connect the last layer to all feature maps. We will continue with the CNN module later. PAN is based on FPN, adding one more layer to the number of concatenated layers and merging instead of adding parts. Head follows YOLOv3 and consists of multiple layers of convolution.

For data enhancement, YOLOv4 uses the new data enhancement method Mosaic. Adopt the method of random zooming and clipping, mix and splice four kinds of pictures for training. This way, we can enrich the training dataset and make the model training more stable and improve the overall training effect. The method Drop block is used instead of regularization. The whole region is deleted randomly. For example, the whole part of the head and heel is deleted. At this time, the neural network will pay attention to learning some features to realize the correct classification.

In addition to good identification accuracy, this study must also have good immediacy. Considering the efficiency of identification hardware, memory use, and other factors will affect the identification speed. The structure of YOLOv4-Tiny is a stripped-down version of YOLOv4, a lightweight model. The parameter is only 6 million, which is one-tenth of the original model. The overall network has 38 layers, and 40.2% AP and 371% FPS were obtained from MS COCO data.

#### 3.1.2. YOLOX

The YOLOX [34] algorithm is improved based on YOLOv3. YOLOX-L obtained 50.0% AP on COCO. On a single Tesla V100 display card, the recognition rate can reach 68.9 FPS. YOLOX-Tiny and YOLOX-nano are 10% better at AP than their YOLOv4-Tiny counterparts. The first change is to replace YOLO’s old Coupled Head with a Decoupled Head. The original Coupled Head predicted the classification, object frame, and regression correction in the same convolutional layer. In the Decoupled Head, classification and regression correction are separated into two convolutions for prediction. Because of two convolutions, the overall computation will be relatively large. However, the feature dimension is reduced to 256 by 1 × 1 convolution and then divided into two branches for classification and regression of the convolution. The classification branch is responsible for classification, and the regression branch predicts the position and size of object boxes and correction boxes. As a result, the overall amount of computation is significantly reduced. Only a few parameters are added, so the overall speed is similar to the original Coupled Head. The overall convergence speed and accuracy are higher than in the previous versions.

#### 3.1.3. SSD

The Single-Shot MultiBox Detector (SSD) [35] was developed in 2016 by Szegedy et al., which belongs to a one-stage object identification method. For an input image, SSD could extract features through convolutional neural networks (CNN), which generate a feature map. The default box is then generated at each point in the feature map, and then all the default boxes are aggregated, followed by the NMS (Non-Maximum Suppression) for most of the bounding boxes. For 300 × 300 image inputs, SSD achieved a FPS of 59 and mAP of 74.3% in the VOC test, and mAP also performed 76.9% for 512 × 512 inputs, outperforming the Faster R-CNN model.

The backbone of SSD is VGG16 in which the Pool5 layer is changed from (size = 2 × 2, Stride = 2) to (size = 3 × 3, Stride = 1), and the last two fully connected layers FC6 and FC7 are replaced by 3 × 3 and 1 × 1 convolution, respectively. The dropout layer and FC8 layer are also removed and four convolution layers are added. SSD adds the Pyramidal Feature Hierarchy, which detects features in different sizes, and references Anchors in Faster R-CNN [36], which sets default boxes with different sizes and aspect ratios on each Feature graph. The prediction boxes refer to these default boxes to make predictions, so the model does not have to learn the boxes’ size, making it easier to train.

### 3.2. System Design and Architecture

This subsection describes the system schematic diagram for rail track inspection, as shown in Figure 1a, which consists of two cameras, a GPS device (Thinkstar technology company, New Taipei City 234, Taiwan (R.O.C.)) with Gyro, a storage device, a laptop PC, and a cloud server. The communication between the local and the cloud server is via a wireless communication module. Use a flat patrol car, as shown in Figure 1b, as the platform loading human operators and all hardware, including a dynamo. Steel frames are used to install and fix three laptop computers and six high-speed cameras, as shown in Figure 1c, along with all wire connectors. The laptop PC (Micro-Star International Co., New Taipei City, Taiwan (R.O.C.)) specs are Intel i9-10980HK CPU, 32 GB DDR4 memory, and Nvidia GeForce RTX 2080 super 8G GPU. The laptop can process two video streams from two cameras without delay. As shown in Figure 1d, LED lights are also fixed under the flat car so we can take a picture of the track for subsequent identification. The GPS with inertial navigation records the position when capturing images. OpenCV API library [37] is adopted to acquire images taken by the high-speed camera and later processing. To utilize the laptop’s full power, the multi-thread parallel processing technique is used to implement the recognition program.

#### 3.2.1. Real-Time Recognition System Diagram

The rail inspection workers must fix the missing components right after spotting them. We must develop the inspection software to process the image right after capturing it. The real-time identification system in this study was performed directly by a high-performance laptop. The identification results are displayed after neural network identification. Considering that the system is run on a flat car traveling at 30 km/h or more and that the track is divided into left and right rails, two GoPro images must be read simultaneously. It was calculated that each image would cover a distance of approximately one meter. The flat car moves approximately eight meters in a second. So eight images per second need to be processed. Moreover, because two cameras are capturing the scene simultaneously, the computer needs to identify 16 or more images at the same time to ensure no missing images. Therefore, it is necessary to select a neural network with good identification accuracy and high FPS as the identification model of the system.

We use multi-process to implement the system program and divide it into three processes: user interface, image processing, and AI recognition, which run simultaneously. Moreover, use the object-oriented way to package the recognition results. Use two queues as a medium for processes to communicate with each other. First, when the program is started, the identification model will be loaded at the same time. Turn on the left and the right camera and display the user interface. The camera will resize all the captured images first, convert them to 416 × 416 and then store them in Queue1. The identification thread will also detect whether there are pictures in Queue1 that need to be identified. If Queue1 is not empty, take it out and identify it. Then, store the result in Queue2. At this time, the user interface will also continuously read whether Queue2 has recognized pictures to display. If Queue2 is not empty, then display detailed identification results, GPS positioning, and other information on the user interface. Finally, when the program is terminated, all recognition results will be exported to an excel sheet for users to view. A detailed system activity diagram is shown in Figure 2.

In order to allow users to operate and view conveniently, this system uses PyQt5 to implement the user interface. First, open the real-time interface system, which will enter the initialization interface. The user can select information such as rail station interval, track type, and lens to record the left and right track images, respectively. After pressing the button on the top view or the side view, it will enter the recognition interface. The screenshot of recognition is shown in Figure 3. Three images will be displayed on the top. The leftmost is the image of the left track, the middle is the image of the right track, and the far right is the image of the missing fastener. Then, there is the current time and the track information just selected. The bottom of the table will list the details of all damaged fasteners. After closing the system, the form will be automatically downloaded as an Excel file. Moreover, store all missing fastener photos for easy viewing by users.

#### 3.2.2. Image Acquisition

In this study, images were captured using area cameras. The camera needs to be set up on the flat maintenance car. The driving speed of the flat maintenance car is approximately 30 km/h, and the distance between consecutive fasteners is approximately 30 cm. To ensure every fastener can be covered by at least one image, the camera needs to capture at least 30 images per second so there will be no missing frames. Therefore, when choosing a camera, the FPS should be at least 30 when capturing images. The most important aspect of image recognition is the clarity of the image. If the image is blurred, it may not be used for classification. Therefore, it is necessary to use a camera that can effectively absorb shocks to avoid unclear images caused by the shaking of the flat car. The weather during the inspection may be rainy. The camera must also have a specific waterproof function so that it can be used usually, whether it is rainy or sunny. The camera must also be connected to the laptop so the program can capture it directly. The camera should not be too complicated to operate so that inspection personnel can use it easily. In addition, the price should not be too high. Based on the above considerations, the GoPro7 and GoPro8 cameras were finally selected.

The GoPro camera is a well-known sports camera in recent years, and its main feature is the ability to record while exercising. This product has a high resolution and excellent hyper smooth super shockproof and waterproof function to a depth of 10 m. GoPro cameras themselves have other features such as built-in Wi-Fi and Bluetooth. This device can be directly used for wireless network transmission, and there is no need to purchase other network modules.

### 3.3. Rail Track Component Classes and Data Collection

The Dajia Branch of the Taiwan Railway Administration provided this study with the track component data, including the image of defective components. Moreover, we discussed with the Dajia branch whether the classification of components meets the status of track inspection operations. Then, we marked and trained the data for this study. Afterward, the Dajia Branch Office provided missing data from time to time and continuously updated the number of data for use in model training. According to the style of camera setup, there are two kinds of datasets: top view and side view.

#### 3.3.1. Top-View Dataset

The top view of the rail track components dataset is divided into ten categories: slightly defective e-clips, severely defective clips, slightly defective spikes, severely defective spikes, slightly defective rail surfaces, severely defective rail surfaces, minor defective track spikes, severely defective track spikes, other missing and shadowed components. The description along with image of defective components is shown in Table 2 [38], where X represents slightly defective, and XX represents severely defective. Sample pictures are shown in Figure 4 and Figure 5.

There are 1417 images in the top view missing construction data and 2380 images in the standard dataset. The total number is 3800 images. In this study, the number of top-view training samples is 3420 images, and the number of test samples is 380. Table 3 shows the distribution of top-view data samples.

#### 3.3.2. Side-View Dataset

The targets of side-view recognition are steel rails and fishplates, and it mainly recognizes areas that the top view cannot see. This study divides the side-view defect categories into the following four categories—weld cracks, rail cracks, fishplate cracks, and fishplate bolts falling off—as shown in Table 4. In addition, regular fishplates and regular welds are included in the training dataset to reduce misjudgments as much as possible. There are 259 images of defective components in the side view, 708 images of regular data, and 967 images total. In this study, the number of side-view training samples is 795, and the number of test samples is 172. The detailed data distribution of each category is shown in Table 5.

#### 3.3.3. Data Augmentation

In order to produce effective training results, a certain amount of data is required for deep learning. This research has particular difficulties in collecting data. Some missing component data are not easy to obtain, such as missing rail surfaces in top-view data and rail cracks in side-view datasets. Therefore, it is necessary to use data augmentation technology to increase the amount of data to prevent the overfitting neural network problem during training.

For the rail surface defect categories in the top-view dataset, Taiwan Railway Dajia Branch can only provide a little information. Therefore, this study uses random flipping, translation, grayscale, and blurring methods to amplify the dataset size. We expand the original picture into four, and try to solve the problem of insufficient data. An example of augmented data is shown in Figure 6.

The Dajia Branch Station can only provide dozens of image data for the rail cracks in the side-view dataset. Welding cracks are entirely unobtainable. The dataset cannot be effectively expanded by conventional augmentation methods such as flipping and translation. Therefore, this study first uses the method of artificial synthesis to remove the background of the wall crack images found on the Internet and then paste them on the normal railway track pictures, as shown in Figure 7. However, artificially compositing images is time-consuming, and generating multiple images at once is impossible. Therefore, this study uses the generative adversarial neural network Cycle GAN [39] to generate rail crack images automatically. The total number of training iterations is 1000, and the time is approximately six hours. The training set consists of 30 images of regular rails and 30 images of artificially synthesized broken rails. Input the pictures of regular rails during the test can automatically generate synthetic rail crack pictures, as shown in Figure 8. The side-view dataset currently has 144 authentic images and 115 artificial images. The detailed distribution of natural and artificial image data of each category is shown in Table 6.

## 4. Experimental Results and Analysis

Section 4.1 presents the training results and comparison of each neural network model in this study in detail. Section 4.2 gives the results of the field test of the real-time identification system of this study to verify its feasibility. The last section carries out the cost calculation of this system.

### 4.1. AI Model Training

The training environment of this study uses a self-assembled personal computer with an AMD Ryzen 7 2700X processor, a GeForce RTX 2080Ti graphics card, and 16G memory. We compare five neural network models, namely YOLOv4, YOLOv4-Tiny, YOLOX-Tiny, SSD512, and SSD300. The neural network model used in the real-time recognition system must have a reasonable recognition rate and recognition speed to avoid frame loss. The reference indicators for comparison are mAP, the recall rate, precision, and FPS.

The evaluation Index of the AI model is mainly focused on the confusion matrix. TP (True Positive) means that the prediction is positive and the ground truth is also true. TN (True Negative (TN) means that the prediction is negative and the ground truth is true. FP (False Positive) means the prediction is positive, but the ground truth is false. FN (False Negative) means that the prediction is negative and the ground truth is positive. Precision is the ratio (TP/TP + FP) that is predicted to be positive and is positive. The recall rate means the ratio (TP/TP + FN) that is positive and is indeed predicted as positive. The mAP is the average of all category Aps. As for AP, it is the area covered by precision and the recall rate under different confidence values (precision-recall curve).

#### 4.1.1. YOLOv4 and YOLOv4-TinyModel Training

In this experiment, the YOLOv4 model is trained with 12,000 epochs. The total time for top-view training was approximately 11 h, and the side-view training took approximately 9 h. The training of YOLOv4-Tiny has 12,000 iterations, the total time for top-view training is approximately 4 h, and the side-view training takes approximately 3 h. The amount of top-view training data used is 3800, and the amount of side-view training data is 967. Before training, it is necessary to mark all the components in the image. This study uses LabelImg [40] data labeling tool, as shown in Figure 9. When marking the defective components, the marking scope will also be extended to the peripheral sleepers and a part of the rail surface.

There are ten types of top-view data in this study. At first, the result of the first version of YOLOv4 training was 82% precision, a 95% recall rate, and 83.16% mAP. The first version of YOLOv4-Tiny training results has a precision of 82%, a recall rate of 96%, and an mAP of 90.77%. Among them, the results of precision and mAP are not ideal, and it was found that the main reason is that there are too many FPs. Most of these FPs are standard components that were misjudged as defective. Therefore, this study added all misjudged test data to the training set during the second version of the training and relabeled these data as background data.

Moreover, review all markup data to avoid markup inconsistencies. Retrain the neural network to reduce the chance of misjudgment. After this adjustment, the mAP of YOLOv4 top-view training results increased to 94.47%, the precision increased to 94%, and the recall rate increased to 91%. YOLOv4-Tiny training results mAP increased to 91.66%, precision increased to 92%, and recall rate was 91%. Among them, the recall rate has slightly decreased because, in the data of the minor defect category of the E buckle clip, the displacement of the buckle clip is less clear. The dataset will be continuously updated to improve this situation. The performance comparison between the first version and the second version for YOLOv4 and YOLOv4-Tiny is shown in Table 7 and Table 8, respectively. It can be seen from the training process that the loss has continued to decline while the mAP has continued to rise. The confusion matrix for each category is shown in Table 9 and Table 10. Figure 10 is an example of the recognized defective components in the top view.

This study’s first version of side-view track component detection by the YOLOv4-Tiny neural network model was trained with the side-view dataset. Only four missing categories are used for training. The training result mAP is 99.00%, precision is 100%, and the recall rate is 99%. The data seems to be good. However, there are too many false positives in the actual test, and almost all the welds in the entire rail are regarded as weld fractures. Therefore, this study added two categories of normal welding and regular fishtail in the second edition of training to reduce the misjudgment of the actual test. The result of YOLOv4 training mAP is 99.24%, precision is 97%, and the recall rate is 95%. The YOLOv4-Tiny training result mAP is 99.16%, precision is 96%, and the recall rate is 94%. The overall mAP has also continued to rise. After the actual test, the misjudgment of the whole video also dropped a lot. The detailed confusion matrix of each category is shown in Table 11 and Table 12. Observing the data in the table, we can find that the number of FPS with fractured welds is still not zero, and these are regular welds that have been misjudged as weld cracks. In the future, these misjudged data will continue to be added back to the training dataset to improve the overall recognition rate as much as possible. In addition, there are multiple FNs at the regular welds, and these FNs are because the side rails will be captured together with the rear rails when shooting. However, the welding image on the rear track is too small to be recognized by the neural network, as shown in Figure 11. Figure 12 shows some examples of side-view defective track component identification.

#### 4.1.2. YOLOX-Tiny Training

For comparison with the YOLOv4-Tiny neural network, the same dataset was used for training. The original dataset is allocated 80% as the training set and the other 20% as the test set. The top-view training iterates 3000 times and takes approximately 72 h in total. The number of iterations of side-view training is 3000 times, and it takes approximately 30 h in total. The overall training time of YOLOX-Tiny is relatively long. The mAP of the top view training is 85.22%, the precision is 83%, and the recall rate is 78%. The result of side-view training is that the mAP is 75.26%, the precision is 89%, and the recall rate is 77%. The comprehensive data of YOLOX-Tiny is not ideal. The detailed confusion matrix of each category is shown in Table 13 and Table 14.

On examining all the incorrectly identified pictures, it was found that FN and FP mostly appear in blurred or rotated and toned pictures, as shown in Figure 13a. Photos of this type account for a relatively small percentage of all datasets. It is inferred that the recognition performance may be degraded due to the lack of such data. Therefore, we redistribute the dataset, setting 90% of the photos as the training set and 10% as the test set. Retrain the model using the same number of iterations. The top-view result of this training was 86.87% mAP, 92% precision, and a 82% recall rate. The mAP of the side-view result is 97.55%, the precision is 99%, and the recall rate is 99%. The overall performance has increased significantly; the comparison is shown in Table 15. Although the YOLOX-Tiny neural network model is relatively low in data, it can effectively identify far and small objects. For example, the side-view rail and the rear rail will be included in the shot. The welds on the rear track can also be identified by YOLOX-Tiny, as shown in Figure 13b.

#### 4.1.3. SSD Training

This study also compares the SSD neural network with the YOLOv4-Tiny neural network. The same dataset is used for training and testing during training. On the training data, SSD adopts the VOC data format. During the initial training, it was found that the training speed was slow, and it was found that the overall training speed was slowed down when the picture was resized. Therefore, before training, this study takes image pre-processing and resizes all images (416 × 416) to reduce the overall training time. The top-view training time SSD512 takes approximately 48 h, the precision is 90%, the mAP is 91.10%, and the recall rate is 89%. The SSD 300 took approximately 24 h, the precision was 93%, the mAP was 91.20%, and the recall rate was 96%. Detailed top-view training results are shown in Table 16.

The side-view training time for SSD512 takes approximately 48 h, the precision is 91%, the recall rate is 95%, and the mAP is 99.98%. The SSD 300 took approximately 24 h, the precision was 92%, the recall rate was 94%, and the mAP was 99.49%. The detailed training results of the side view are shown in Table 17.

#### 4.1.4. Comparisons of the Neural Models

Table 18 compares the results of top-view training, and Table 19 compares the results of side-view training. From the perspective of the recall rate, SSD300 will be a better choice. From the perspective of precision and mAP, YOLOv4 is the better choice. However, from the comparison table, it can be found that YOLOv4-Tiny is not inferior to other neural network models. In the side view training results, the precision of YOLOv4-Tiny is 7% higher than that of SSD512. In addition, this research advocates a real-time identification system, so in addition to considering various data, the identification speed (FPS) is also essential.

Based on the above data, YOLOv4-Tiny has an excellent performance in various data indicators. Moreover, the recognition speed is 3~7-fold higher than other neural networks. Therefore, this research considers speed without losing accuracy and finally chooses YOLOv4-Tiny as the neural network model used in the real-time identification system of this research.

### 4.2. Field Test Results

This study conducts field AI detection of the rail track component on the East–West Main Line between Taichung Port Station and Qingshui Station and conducts mutual verification with human detection. First, the defective component needs to be fabricated. Due to the difficulty in producing defective components, it is impossible to produce all defect categories. Therefore, the top-view defect components are mainly the severe defect of the e-clip, while the side-view defect components are mainly the cracks of the rail and the falling off of the fishplate bolts. Moreover, because it is impossible to create natural rail cracks, the cracks on the rail side of the rail were simulated with a marker pen, aluminum tape, and black tape. The simulated crack images were discussed with the personnel of Dajia Railway Station in advance, and they are all similar to the actual rail crack images, as shown in Figure 14. A total of 32 artificial markings were made by hand, including 13 severely missing(XX) e-Clips, one fishtail plate bolt falling off, and 18 simulated rail cracks.

A total of 13 top-view defective components were artificially fabricated. AI detected 14, and the detection rate was 14/13 = 107%. Manual foot inspections found 15. Although AI patrol inspection is less than manual foot inspection by one, the overall time and human resources spent are relatively much less. In the future, we will continue to increase the dataset to reduce misjudgment. The detailed fault track component detection is shown in Table 20.

A total of 19 side-view defect track components were manually fabricated. The AI detected eight, and the human visual inspection by walking detected five. The defective components that AI cannot detect are all images of simulated broken rails on, such as rail head, rail belly, and rail bottom, as shown in Figure 15. The failure to detect is because the features of these simulated broken tracks are too small, so the images of these broken tracks cannot be captured. Because of this, AI cannot detect it correctly. AI can correctly detect other prominent simulated broken rails. The overall AI detection is also three more than the manual detection on foot. The details are shown in Table 21.

### 4.3. Robustness Test

To validate the robustness of the proposed defective rail track classification system, we collected another track dataset of mass rapid transportation that differed from the previously used dataset. The total number of images in the training dataset was 528, and the number of defect categories was five. During the training stage, YOLOv4 took approximately 10 h for 12,000 iterations, while YOLOv4-Tiny took approximately 3 h for the same number of iterations.

After testing, as shown in Table 22, we obtained a 97% precision rate, a 96% recall rate, and 97.3% mAP for YOLOv4. On the other hand, we obtained a 98% precision rate, a 94% recall rate, and 97.5% mAP for YOLOv4-Tiny. Although the mAP and precision rate of YOLOv4-Tiny are good, the recall rate is lower than for YOLOv4. The reason is that there are too many false negatives, and there is a slight overfitting phenomenon. In this test, we successfully demonstrated that the proposed methodology using YOLOv4-Tiny is quite robust. The performance indexes are all over 90%, which could be deployed for various track applications.

## 5. Conclusions

In this study, five neural network models were compared, namely YOLOv4, YOLOv4-Tiny, YOLOX-Tiny, SSD512, and SSD300. After considering precision, the recall rate, mAP, and FPS, YOLOv4-Tiny was finally selected as the neural network model used in this real-time recognition system. Since the railway track is not open to the outside world, it is challenging to collect damaged samples, which also leads to insufficient data for some defect categories in training. In this regard, this study uses data augmentation methods to increase the diversity of data as much as possible using rotation and grayscale. In the side-view data, Cycle Gan is also used to automatically generate images of track cracks to avoid overfitting as much as possible due to insufficient training data.

This system uses a GoPro camera to capture images. We used the multi-process method to write programs and execute three processes concurrently. The recognition results are packaged object oriented so that the system can quickly access data. Combined with the flat maintenance car, real-time identification can be performed at approximately 30 km/h. Moreover, inertial navigation can locate the position of defective components. This system will integrate all the identification results into an excel report and record the storage location of the identification results, 100 m piles, and photos of defective components in detail so that users can view them in real time. While maintaining a reasonable recognition rate, the efficiency of track inspection is greatly improved. Furthermore, it can have lower costs and a safer maintenance environment compared with manual inspection.

Collecting the image data of track defects required in this study is difficult. Although multiple defective component data have been generated through methods such as data enhancement or Cycle GAN, the same photo is still used for rotation and grayscale. There is a slight improvement in overall data richness. In the future, we plan to cooperate with the Taiwan Railway branches in various counties and cities to obtain more missing component data to increase the dataset required for training.

The inertial navigation module used in this research can work in a short tunnel within 500 m, and the positioning error can be less than 10 m. However, since the inertial navigation error will accumulate over time, the positioning error will also increase when encountering a long tunnel. In the future, we hope to find an inertial navigation module with automatic correction or use the track map information provided by the Taiwan Railway Administration to perform positioning correction. This system directly uses a high-performance notebook computer for real-time identification. We want to use lighter neural network models and embedded systems, such as TX2 or Xavier, for real-time recognition. This progress can make the process more convenient to users.

## Figures and Tables

**Figure 1 sensors-22-09970-f001:**
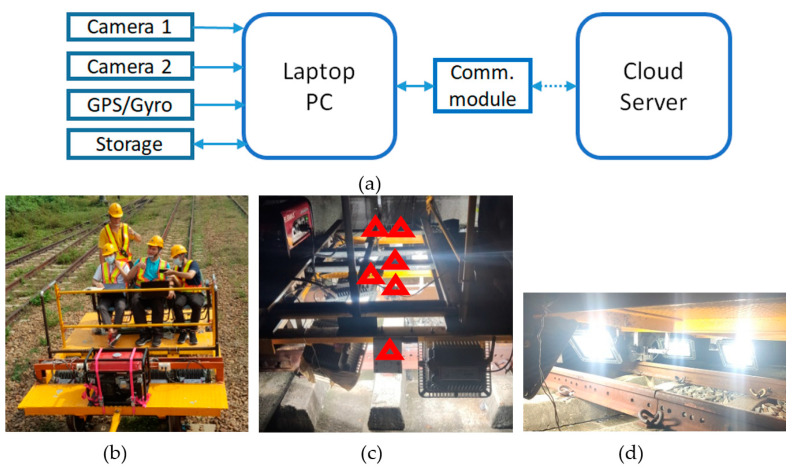
System architecture. (**a**) system schematic diagram; (**b**) a flat patrol car; (**c**) the red triangles are the positions of the cameras; (**d**) LED lights are used for the illumination.

**Figure 2 sensors-22-09970-f002:**
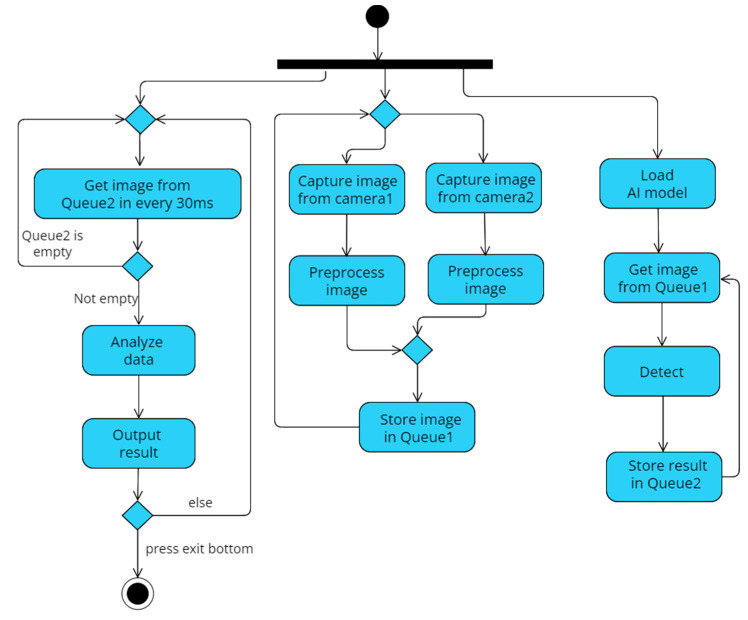
Software activity diagram.

**Figure 3 sensors-22-09970-f003:**
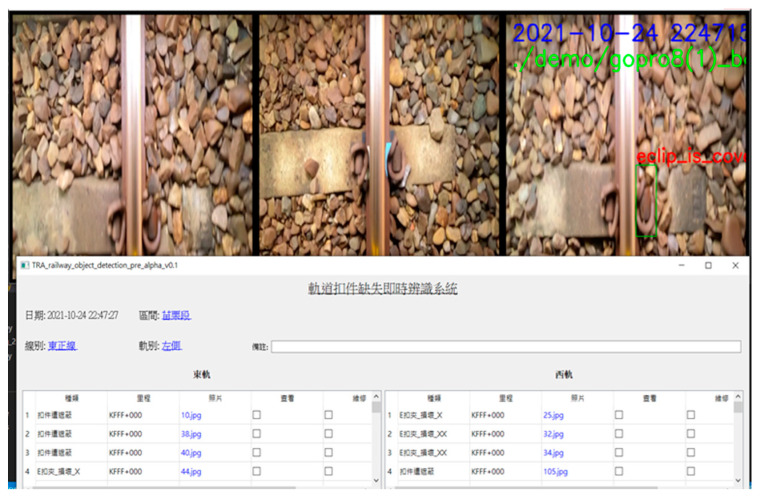
The interface of the track fault detection system.

**Figure 4 sensors-22-09970-f004:**
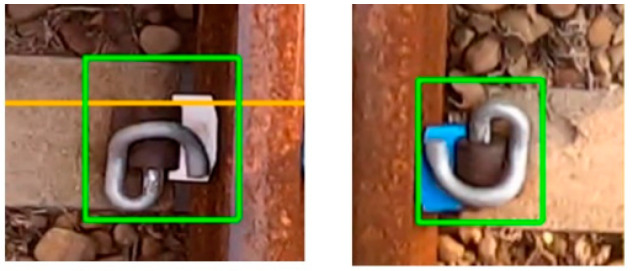
Two samples of light loose (X) track fasteners.

**Figure 5 sensors-22-09970-f005:**
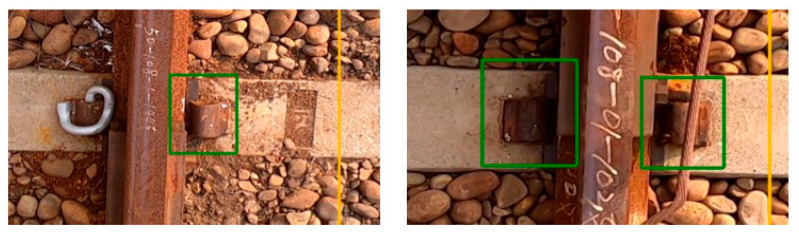
Three samples of severely defective (XX) track components where the fasteners are missing.

**Figure 6 sensors-22-09970-f006:**
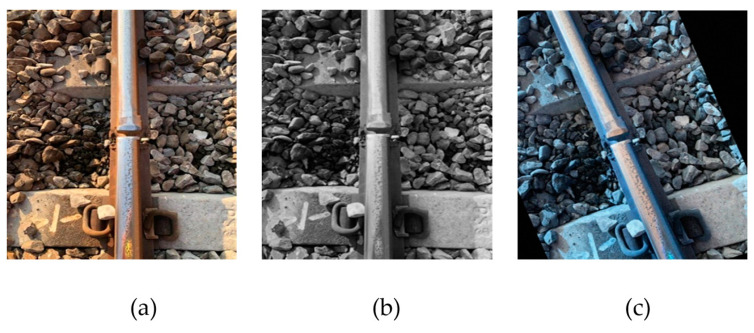
Examples of data augmentation. (**a**) original; (**b**) gray; (**c**) rotation.

**Figure 7 sensors-22-09970-f007:**
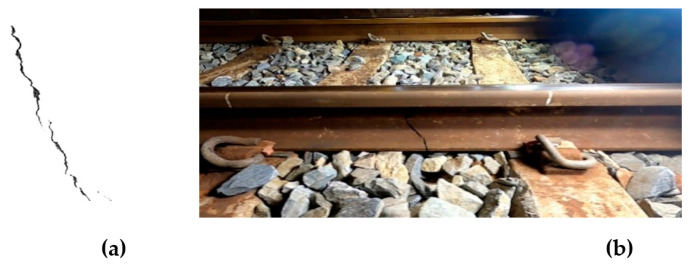
Human synthesized images. (**a**) crack image; (**b**) human-made images.

**Figure 8 sensors-22-09970-f008:**
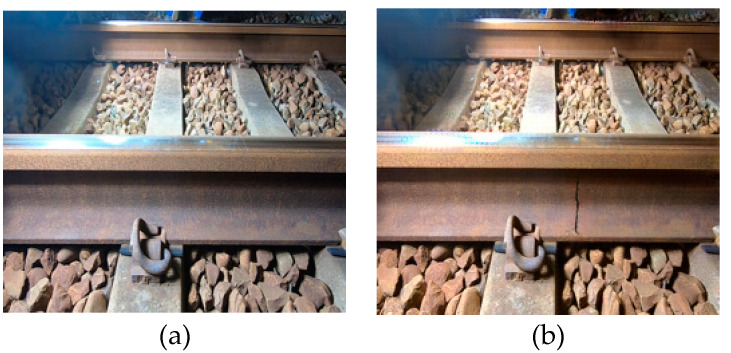
Automatic image generation. (**a**) original image; (**b**) cycle GAN-generated images.

**Figure 9 sensors-22-09970-f009:**
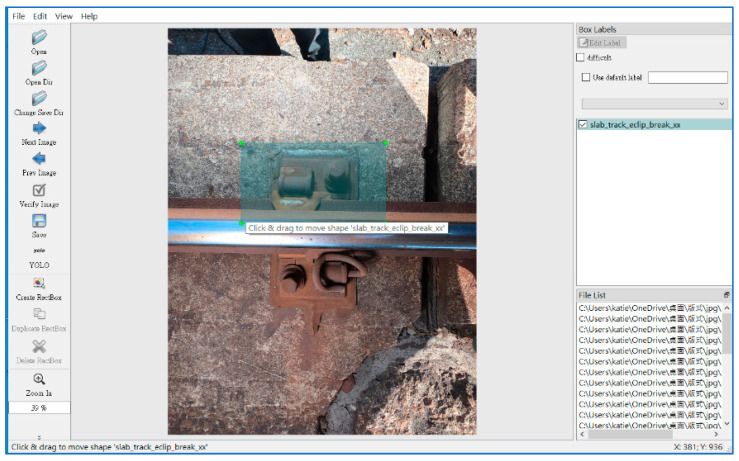
The tool LabelImg is used to create the training dataset.

**Figure 10 sensors-22-09970-f010:**
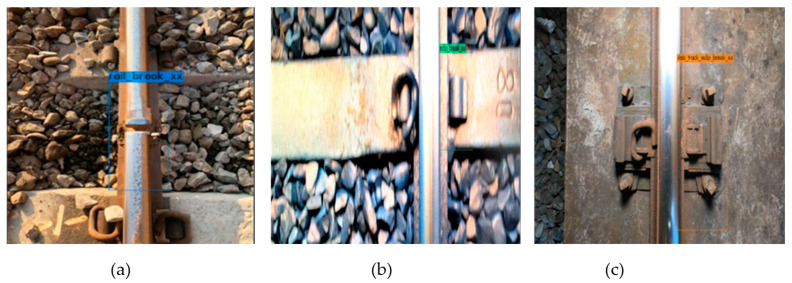
Examples of the detected defective track components in the top view. (**a**) rail surface_XX; (**b**) e_Clip XX; (**c**) slab e_Clip XX.

**Figure 11 sensors-22-09970-f011:**
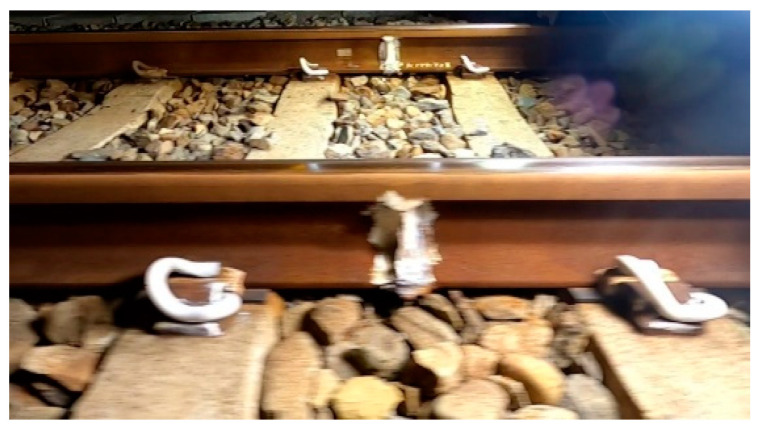
A FN example of Rail_welding_XX.

**Figure 12 sensors-22-09970-f012:**
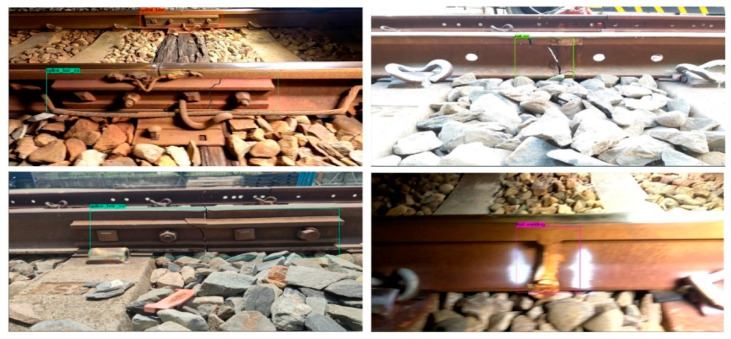
Some examples of detected defective track components in the side view by the trained YOLOv4-Tiny. Upper left: Fishplate_bar_XX (generated); upper right: Rail_XX (real); lower left: Fishplate_bar XX (real); lower right: Rail_welding_O.

**Figure 13 sensors-22-09970-f013:**
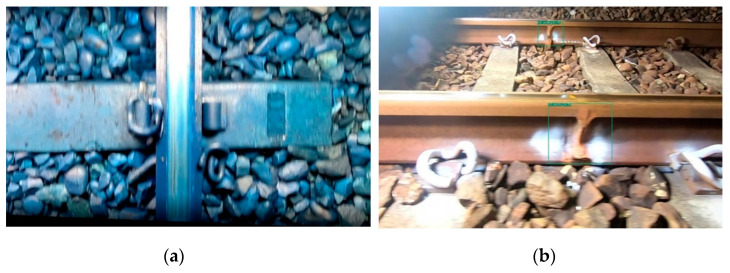
Some false examples. (**a**) too blur to identify; (**b**) multiple detection of Rail_welding_O.

**Figure 14 sensors-22-09970-f014:**
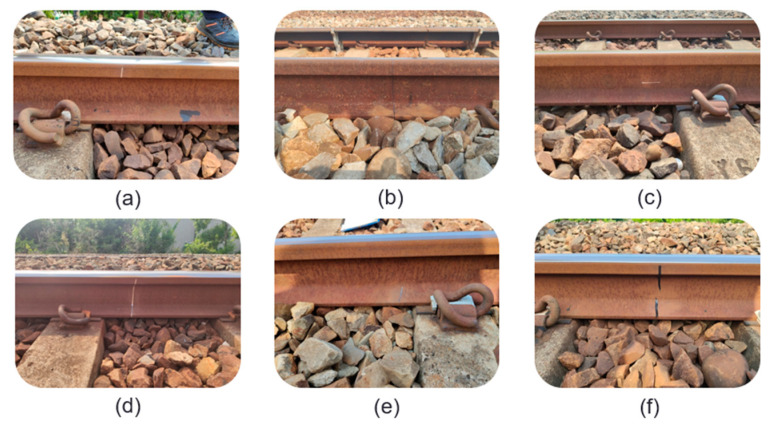
Simulated rail cracks by using (**a**) aluminum tape on rail head, (**b**) marker pen to draw, (**c**) aluminum tape on rail belly, (**d**) aluminum tape, (**e**) aluminum tape on rail bottom, and (**f**) black tape.

**Figure 15 sensors-22-09970-f015:**
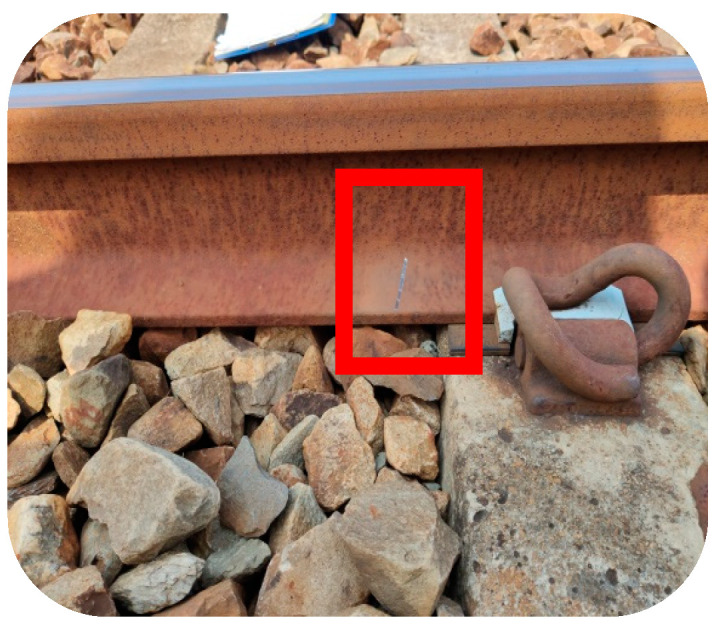
Simulated rail break on bottom in side-view.

**Table 1 sensors-22-09970-t001:** Some commercial track inspection products.

Product	Items	Imaging	Speed
Plasser and Theurer [2]	Geometric alignment, gauge, height level, direction, planarity	Laser scan camera	>100 km/h
Ensco [3]	Fishplate, sleeper, fastener, rail surface, power line	Line/area camera	High speed
Fugro [4]	Overhead line electrification, rail head wear, gauge, tilt, twist, level calculation	Laser scan camera	High speed
JC Tech [5]	Track imaging	4K area camera	<70 km/h
Mermec [6]	Rolling surface, fastenings, sleepers and track bed	Line scan camera	<200 km/h

**Table 2 sensors-22-09970-t002:** Types of the defective track components from the top view [38].

No.	Types	Sample Images	Description
1	e-Clip_X	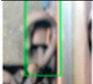	Fastener deviated or not positioned
2	e-Clip_XX	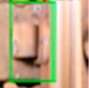	Fastener is lost or broken
3	Spike_X	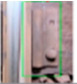	The upper spike is deviated or not positioned
4	Spike_XX	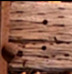	Two spikes are lost or broken
5	Rail surface_X	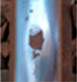	Depression in the center of the rail surface, and slight damage to the rail surface.
6	Rail surface_XX	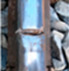	Rail break on top of rail surface
7	Slab e-Clip_X	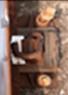	e-Clipper position deviated or bolt loose
8	Slab e-Clip_XX	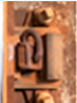	Missing or broken e-Clip or bolts
9	Covered	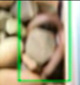	Fasteners are obscured by lines, ballasts, plants, etc.
10	Others	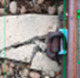	Other foreign objects, garbage on the track

**Table 3 sensors-22-09970-t003:** Top-view dataset.

No.	Types	Real Images	Count (Frame)
1	e-Clip_X	52	52
2	e-Clip_XX	276	276
3	Spike_X	37	37
4	Spike_XX	58	58
5	Rail surface_X	77	77
6	Rail surface_XX	226	226
7	Slab e-Clip_X	35	35
8	Slab e-Clip_XX	28	28
9	Covered	17	17
10	Others	611	611
11	Normal	2383	2383
	Total		3800

**Table 4 sensors-22-09970-t004:** Types of defective track components from the side view.

No.	Types	Sample Images	Description
1	Weld fracture	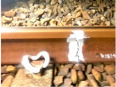	Cracks appearing on rail side welds
2	Rail crack	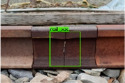	Cracks appearing on the side of the rail
3	Fishtail plate fracture	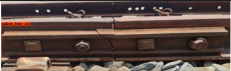	Cracks appearing on the fishplate
4	Fishplate bolts fall off	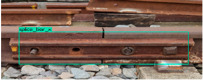	Fish plate bolts are loose or fall off

**Table 5 sensors-22-09970-t005:** The original side-view dataset.

No.	Types	Real Images	Count (Frame)
Rail_welding_XX	20	7	27
Rail_XX	70	38	108
Fishplate_bar_XX	25	12	37
Fishplate_bar_X	61	26	87
Fishplate_bar_O	257	34	291
Rail_welding_O	362	55	417
Total	795	172	967

**Table 6 sensors-22-09970-t006:** The augmented side-view dataset.

Type	Real Image (Frame)	Human Synthesized	Subtotal
Rail_welding_XX	0	27	27
Rail_XX	42	66	108
Fishplate_bar_XX	15	22	37
Fishplate_bar_X	87	0	87
Total	144	115	259

**Table 7 sensors-22-09970-t007:** Performance of the 1st and the 2nd version of YOLOv4.

	YOLOv4 (1st)	YOLOv4 (2nd)
mAP	83.16%	94.47%
Precision	82%	94%
Recall	95%	91%

**Table 8 sensors-22-09970-t008:** Performance of the 1st and the 2nd version of YOLOv4-Tiny.

	YOLOv4-Tiny (1st)	YOLOv4-Tiny (2nd)
mAP	90.77%	91.66%
Precision	82%	92%
Recall	96%	91%

**Table 9 sensors-22-09970-t009:** YOLOv4 training results for each defective type in the top view.

Type	TP	FP	FN
e-Clip_X	9	2	5
e-Clip_XX	68	4	3
Spike_X	36	0	2
Spike_XX	16	2	0
Rail surface_X	25	2	0
Rail surface_XX	25	4	3
Slab e-Clip_X	8	1	1
Slab e-Clip_XX	11	2	0
Other	5	0	0
Covered	133	6	18
Total	336	22	33

**Table 10 sensors-22-09970-t010:** YOLOv4-Tiny training results for each defective type in the top view.

Type	TP	FP	FN
e-Clip_X	8	2	6
e-Clip_XX	63	3	8
Spike_X	35	1	3
Spike_XX	16	3	0
Rail surface_X	24	6	2
Rail surface_XX	27	1	1
Slab e-Clip_X	8	1	1
Slab e-Clip_XX	11	2	0
Other	5	0	0
Covered	138	11	13
Total	335	30	34

**Table 11 sensors-22-09970-t011:** YOLOv4 training results for each defective type in the side view.

Type	TP	FP	FN
Rail_XX	36	3	2
Rail_welding_XX	7	0	0
Rail_welding_O	77	0	8
Fishplate_bar_XX	12	2	0
Fishplate_bar_X	25	0	1
Fishplate_bar_O	49	1	0
Total	206	6	11

**Table 12 sensors-22-09970-t012:** YOLOv4-Tiny training results for each defective type in the side view.

Type	TP	FP	FN
Rail_XX	37	0	1
Rail_welding_XX	7	9	0
Rail_welding_O	73	0	12
Fishplate_bar_XX	12	0	0
Fishplate_bar_X	26	0	0
Fishplate_bar_O	49	0	0
Total	204	9	13

**Table 13 sensors-22-09970-t013:** YOLOX-Tiny training results for each defective type in the top view.

Type	TP	FP	FN
e-Clip_X	6	10	1
e-Clip_XX	36	4	15
Spike_X	29	0	3
Spike_XX	12	2	1
Rail surface_X	16	0	0
Rail surface_XX	32	4	1
Slab e-Clip_X	12	8	0
Slab e-Clip_XX	17	0	3
Other	2	0	1
Covered	97	24	49
Total	259	52	74

**Table 14 sensors-22-09970-t014:** YOLOX-Tiny training results for each defective type in the side view.

Type	TP	FP	FN
Rail_XX	17	1	1
Rail_welding_XX	4	0	1
Rail_welding_O	45	0	24
Fishplate_bar_XX	4	2	1
Fishplate_bar_X	11	0	1
Fishplate_bar_O	46	12	10
Total	127	15	38

**Table 15 sensors-22-09970-t015:** YOLOX-Tiny re-training performance in the top view and the side view.

	Top View		Side View	
	8:2	9:1	8:2	9:1
mAP	85.22%	86.87%	75.26%	97.55%
Precision	83%	92%	89%	99%
Recall	78%	82%	77%	99%

**Table 16 sensors-22-09970-t016:** SSD 512/300 training results of the top view.

Type	TP	FP	FN
e-Clip_X	3/7	2/0	4/0
e-Clip_XX	51/51	3/4	1/1
Spike_X	20/20	2/0	2/0
Spike_XX	10/9	3/4	5/4
Rail surface_X	17/23	2/0	11/5
Rail surface_XX	39/49	5/1	11/0
Slab e-Clip_X	13/13	0/0	0/0
Slab e-Clip_XX	19/20	5/2	4/3
Other	13/12	1/0	1/2
Covered	150/150	15/16	4/0
Total	335/356	38/27	43/15

**Table 17 sensors-22-09970-t017:** SSD512/300 training results of the side view.

Type	TP	FP	FN
Rail_XX	37/36	4/3	1/2
Rail_welding_XX	7/7	10/9	0/0
Rail_welding_O	82/82	1/1	1/1
Fishplate_bar_XX	12/11	0/0	6/7
Fishplate_bar_X	25/25	0/0	1/1
Fishplate_bar_O	49/49	6/6	2/2
Total	212/209	21/19	11/13

**Table 18 sensors-22-09970-t018:** Comparisons of the top-view training results among different neural models.

	YOLOv4-Tiny	YOLOv4	YOLOX-Tiny	SSD512	SSD300
mAP	91.66%	94.47%	85.22	91.10%	91.20%
Precision	92%	94%	83%	90%	91%
Recall	91%	91%	78%	89%	95%
FPS	150	20~22	50~55	22~24	45~47

**Table 19 sensors-22-09970-t019:** Comparisons of the side-view training results among different neural models.

	YOLOv4-Tiny	YOLOv4	YOLOX-Tiny	SSD512	SSD300
mAP	99.16%	99.24%	75.26%	99.98%	99.49%
Precision	96%	97%	89%	93%	92%
Recall	94%	95%	77%	96%	94%
FPS	150	20~22	50~55	22~24	45~47

**Table 20 sensors-22-09970-t020:** Top-view defect inspection by AI and human.

	Man Made Markers	Human Visual Inspection	AI Inspection
Top-view defects	13	15	14
Recall rate		115%	107%

**Table 21 sensors-22-09970-t021:** Side-view defect inspection by AI and human.

	Man Made Markers	Human Visual Inspection	AI Inspection
Side-view defects	Total: 19, (9 if 10 short tapes on track belly are not counted)	5	5 (aluminum tape) 1 (black tape) 1 (marker pen) 1 (fishplate)
Recall rate		26.3%	42.1%

**Table 22 sensors-22-09970-t022:** YOLOv4 and YOLOv4-Tiny training and testing results for a different dataset.

	YOLOv4	YOLOv4-Tiny
mAP	97.3%	97.5%
Precision	97%	98%
Recall	96%	94%

## Data Availability

Not applicable.

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
