# Peer review of "An Online Rail Track Fastener Classification System Based on YOLO Models"

_sensors, 2022, doi:10.3390/s22249970_

Round 1

Reviewer 1 Report

Dear Authors,

technical inspection of railway tracks is a very important element of the safety system in rail transport. The authors use automation and vision techniques of computer image processing to analyze the quality of railway rails. The manuscript proposes the use of the YOLOv4-Tiny neural network to identify railroad track damage in real time. Standard damage occurring in railway tracks is analysed. The authors extensively presented the results of research and experiments, the work contains a rich illustrative contribution. The data is collected and presented clearly in the form of tables. In general, the work is well written, it is extensive (25 pages), but there are some minor shortcomings, mainly editing.

First, in my opinion abstract should be shortened to the recommended 200 words.

Secondly, the cameras used in the research are described too generally, only the type is given. I think it is worth revealing more technical details, camera manufacturer, resolution, accuracy, etc.

Third, there are minor editing errors, which are as follows:

Some editing errors for example: line 170. Similarly in the description of Figure 6 and 7, lines 430 and 449.

Line 324 you write the size of the images is 416*416, while in another work line 595 is the record (416x416), please standardize it.

Other errors, lines: 552, 553.

Fourth, the reference list should be adapted to the journal template.

To sum up, I believe that the work is suitable for printing after minor deficiencies have been corrected.

Thank you

Author Response

Reviewer 1 Comments:

Technical inspection of railway tracks is a very important element of the safety system in rail transport. The authors use automation and vision techniques of computer image processing to analyze the quality of railway rails. The manuscript proposes the use of the YOLOv4-Tiny neural network to identify railroad track damage in real time. Standard damage occurring in railway tracks is analysed. The authors extensively presented the results of research and experiments, the work contains a rich illustrative contribution. The data is collected and presented clearly in the form of tables. In general, the work is well written, it is extensive (25 pages), but there are some minor shortcomings, mainly editing.

  1. First, in my opinion abstract should be shortened to the recommended 200 words.

Response 1: Thanks for the suggestion and the abstract is shortened no more than 200 words.

  1. Secondly, the cameras used in the research are described too generally, only the type is given. I think it is worth revealing more technical details, camera manufacturer, resolution, accuracy, etc.

Response 2: Thanks for the valuable suggestion. The required specifications of the camera, such as resolution, fps, vibration, waterproof, and manufacturer, are all described in Sec. 3.2.2.

  1. Third, there are minor editing errors, which are as follows:

Some editing errors for example: line 170. Similarly, in the description of Figure 6 and 7, lines 430 and 449.

Line 324 you write the size of the images is 416*416, while in another work line 595 is the record (416x416), please standardize it.

Other errors, lines: 552, 553.

Response 3: Thanks for the valuable suggestions. All the editing errors are corrected.

  1. Fourth, the reference list should be adapted to the journal template.

Response 4: Thanks for the valuable suggestion. The reference parts are modified to follow the journal template.

  1. To sum up, I believe that the work is suitable for printing after minor deficiencies have been corrected.

Response 5: Your positive comments are highly appreciated.

Reviewer 2 Report

The manuscript proposed real-time track defective inspection system includes a high-performance notebook, two sports cameras, six LED project lights, three parallel processes, and object-oriented packaged results. The whole system is mounted on a flat cart running at 30 km/hr, and our system can process the captured images by the two sports cameras in real time. Several details should be illustrated as the following:

1.     The schematic diagram of the proposed algorithm should be depicted in the manuscript for rail track fasteners classification?

2.     Types of the defective track components in Table 2 should be illustrated in details.

3.     Classifying results should be discussed and compared to other algorithms or system.

4.     The English requires more attentions throughout the manuscript. The authors should check the English presentation throughout the manuscript.

Author Response

Reviewer 2 Comments:

The manuscript proposed real-time track defective inspection system includes a high-performance notebook, two sports cameras, six LED project lights, three parallel processes, and object-oriented packaged results. The whole system is mounted on a flat cart running at 30 km/hr, and our system can process the captured images by the two sports cameras in real time. Several details should be illustrated as the following:

  1. The schematic diagram of the proposed algorithm should be depicted in the manuscript for rail track fasteners classification?

Response 1: Thanks for the valuable comment. We added Fig. 1(a) to show the system schematic diagram which consists of two cameras, a GPS device with Gyro, a storage device, a portable PC, and a cloud server. The communication between the local and the cloud server is via a wireless communication module.

  1. Types of the defective track components in Table 2 should be illustrated in details.

Response 2: Thanks for the valuable comment. More description of the defectve types are added in Table 2.

  1. Classifying results should be discussed and compared to other algorithms or system.

Response 3: Thanks for the valuable comment. We summarized the test results and gave Tables 18 and 19 for the comparisons with some state-of-the-art deep learning neural network models. Based on the performance index, YOLOv4-Tiny has excellent performance and 3~7 times more recognition speed than other neural network models. Therefore, we chose YOLOv4-Tiny by considering speed without losing accuracy.

  1. The English requires more attentions throughout the manuscript. The authors should check the English presentation throughout the manuscript.

Response 4: Thanks for the valuable comment. The manuscript is checked carefully word by word to ensure no typos and grammar errors.

Reviewer 3 Report

This is an interesting manuscript and the authors have done a very good to represent the topic and the methodology. The paper is generally well written and structured. However, in my opinion the paper has some shortcomings  regards to the data analyses , and I feel this this study will more valuable if it used more than one dataset.

Author Response

Reviewer 3 Comments:

This is an interesting manuscript and the authors have done a very good to represent the topic and the methodology. The paper is generally well written and structured. However, in my opinion the paper has some shortcomings regards to the data analyses, and I feel this study will more valuable if it used more than one dataset.

Response 1: Thanks for the valuable comments. We replace the original heading of Section 4.3 for cost analysis with the new heading “Robustness Test” for another dataset testing. Compared with the previously used dataset, the proposed methodology using YOLOv4-tiny is quite robust. The performance indexes are all over 90% which could be deployed for various track applications to save human efforts.

Round 2

Reviewer 2 Report

The manuscript is well revised, and it could be accepted for publishication in the present form.